

# How many reptile and amphibian species are in Uganda, and why it matters for global biodiversity conservation

Daniel F. Hughes[1] and Mathias Behangana[2]

[1] Department of Biology, Coe College, Cedar Rapids, IA, United States of America
[2] NICE Planet Uganda Limited, Kampala, Uganda

## ABSTRACT

Biodiversity is unevenly distributed across the globe. Regional differences in biodiversity impact conservation through the allocation of financial resources, development of infrastructure, and public attention. Such resources are often prioritized to areas that are in more need than others. However, reasons for deciding which locations are more deserving are derived from an accurate knowledge of the number and composition of species that occur in each region. Regional differences in biodiversity, however, can reflect differences in the source of information consulted, rather than *bona fide* differences between areas. As a result, conservation resources may not be directed to regions in proportion to their actual need, especially if there is no consensus among sources of information. Here, we compared major sources of information on species of reptiles and amphibians that occur in Uganda, Africa. We found that none of the sources agreed on the total number, nor composition, of species in the country, with estimates for amphibians ranging more widely than those for reptiles. Notably, sources with similar species richness differed in species composition, which had an impact on the number of threatened species in the country. These results for a conspicuous group of vertebrates suggest that lesser-known groups are also likely misrepresented in sources, especially in other underexplored regions of tropical Africa. We discuss the implications for biodiversity conservation that are derived from inaccurate species lists that are commonly used by agencies, scientists, and practitioners. We argue that more critical evaluations of biodiversity resources, in addition to greater capacity building for field programs, taxonomy, and museum collections, will be essential to ensure that resources are allocated to regions that need them the most.

# INTRODUCTION

Biodiversity hotspots are characterized by an exceptionally high amount of unique biological diversity within relatively small geographic areas where the perceived threat level is disproportionately high compared to other areas (*Reid, 1998*). Consequently, organizations, governments, and agencies often prioritize hotspots to receive more conservation resources and attention due to the immediate risk of losing unique species and ecosystems (*Leader-Williams & Albon, 1988*; *Wilson et al., 2006*; *Marchese, 2015*).

Corresponding author
Daniel F. Hughes, dhughes@coe.edu

Dedicating such resources to these regions is considered an optimal strategy for preserving the most amount of Earth's biodiversity in the shortest time (*Mittermeier et al., 2011*; *Kareiva & Kareiva, 2017*). Many global initiatives, such as the Convention on Biological Diversity, also recognize the significance of hotspot preservation in achieving broader conservation objectives and conservation targets (*Chandra & Idrisova, 2011*). As a result, funding agencies and donors are more inclined to support initiatives that contribute to these global targets, making hotspots attractive recipients for financial resources (*Waldron et al., 2013*). However, the provisioning of resources is not always proportionate to the level of need, in part, because an uneven distribution of biodiversity knowledge (*Hortal et al., 2015*) produces misallocations where deserving regions may miss out on crucial resources (*Reed et al., 2020*; *Chapman et al., 2024*).

The determination of the number of species in a region, such as a biodiversity hotspot, relies on taxonomic inventories (*Myers, 1990*), but the accuracy of species lists is influenced by the extent of research conducted (*Fisher et al., 2011*; *Titley, Snaddon & Turner, 2017*). Some regions, often less explored or lacking adequate research infrastructure, suffer from a paucity of scientific investigations into their biodiversity (*Etard, Morrill & Newbold, 2020*; *Feng et al., 2022*). Consequently, the number of species reported for such regions may be misleadingly low, creating a skewed representation of its biodiversity. Tropical Africa exemplifies this issue as many countries lack comprehensive knowledge of their biodiversity (*Tolley et al., 2016*; *Stephenson et al., 2017*; *Siddig, 2019*; *Achieng et al., 2023*), leading to overlooked species, misidentifications, and gaps in understanding the true extent of biological diversity in the continent (*Gaston & O'Neill, 2004*; *Hochkirch et al., 2021*).

Repositories that synthesize biodiversity information, including internet platforms and synoptic guides, play pivotal roles in providing accessible data to conservation practitioners on species in different regions (*Smith et al., 2000*; *Vanden Eynden, Oatham & Johnson, 2008*; *Robertson et al., 2014*). These resources contribute to decision-making processes in conservation, helping to summarize our collective knowledge in an effort to identify regions that require the most attention and resources, which is based largely on the number of species and their threats. The International Union for Conservation of Nature Red List of Threatened Species, for example, is a comprehensive online database that provides assessments of species' extinction risk, representing a valuable tool for prioritizing conservation efforts (*Rondinini et al., 2014*). However, such biodiversity resources are assumed to be based on highly accurate and up-to-date information (*Stephenson & Stengel, 2020*). Inaccurate species numbers and composition, often stemming from incomplete data, can result in a biased representation of biodiversity (*Cantú-Salazar & Gaston, 2013*; *Hughes et al., 2021*). Errors may arise from misidentifications, outdated information, or gaps in research efforts, particularly in regions where biodiversity knowledge is limited, such as in the Afrotropics (*Tolley et al., 2016*; *Ficetola et al., 2014*; *Feldman et al., 2021*). Consequently, conservation funds may be misallocated based on faulty assessments, potentially neglecting areas with high levels of unrecognized biodiversity value. To mitigate the risk of misallocation, it is crucial to critically evaluate and assess repositories for biodiversity information before implementation, but few conservation practitioners and agencies have enough time to do such due diligence.

Here, we use information on the reptiles and amphibians of Uganda, Africa, available in popular repositories as a microcosm of a potentially widespread issue among taxonomic groups in underexplored regions. We critically assessed major sources for information on reptiles and amphibians in this tropical African country, which revealed a glaring lack of consensus on the total number of species, composition, or the proportion of threatened species. The discrepancies in taxonomic knowledge we observed across several biodiversity sources highlight the potential pitfalls of relying on incomplete or outdated species lists to determine conservation priorities, which could also affect global analyses of biodiversity and biogeography. Our work underscores the notion that certain groups, such as small nondescript amphibians, are most at risk for being underrecognized. Consequently, resources should be allocated for phylogenetic studies aimed at delimiting poorly resolved species, increased frequency of species assessments, and capacity building in biodiverse countries. The implications of our findings extend well beyond Uganda, and we argue should serve as a cautionary tale for conservation decision-makers in global biodiversity.

## METHODS

On 26 December 2023, we systematically gathered species lists from major biodiversity repositories, academic databases, and comprehensive books on the herpetofauna of the East African country of Uganda (Fig. 1). We focused on comprehensive, secondary sources because they represent our collective knowledge on what occurs in the country and are commonly referenced for regional assessments of biodiversity. We compared species lists of amphibians and reptiles in Uganda among popular biodiversity resources: International Union for Conservation of Nature (IUCN) Red List of Threatened Species (https://www.iucnredlist.org/), iNaturalist (https://www.inaturalist.org/), Global Biodiversity Information Facility (GBIF) (https://www.gbif.org/), iDigBio (https://www.idigbio.org/), and VertNet (http://www.vertnet.org/index.html). We also compared major sources that were specific to amphibians (AmphibiaWeb (https://amphibiaweb.org/), Amphibian Species of the World (ASW) (https://amphibiansoftheworld.amnh.org/), and *Channing & Rödel (2019)*) and reptiles (Reptile Database http://www.reptile-database.org/), RepFocus (https://www.repfocus.dk/index.html, and *Spawls et al. (2018)*). We consulted a range of sources with varying source material in an attempt to capture the most complete assessment of the herpetofauna of this country. We note that sources that relied on user observations, such as iNaturalist, were not expected to be comprehensive but we included them in analyses because they are the most popular resources for biodiversity knowledge available online today.

For books, we examined range maps in consultation with the text provided for each species to determine if it occurred in Uganda. For websites, we restricted the locality to "Uganda" in the search tools and downloaded the list of species that were purported to occur there as according to each source. We examined all lists closely and obvious taxonomic discrepancies within each list were addressed through a standardized procedure. We used the most up-to-date academic websites for each group (ASW for amphibians and The Reptile Database for reptiles) to reconcile various nomenclatural issues within each list,

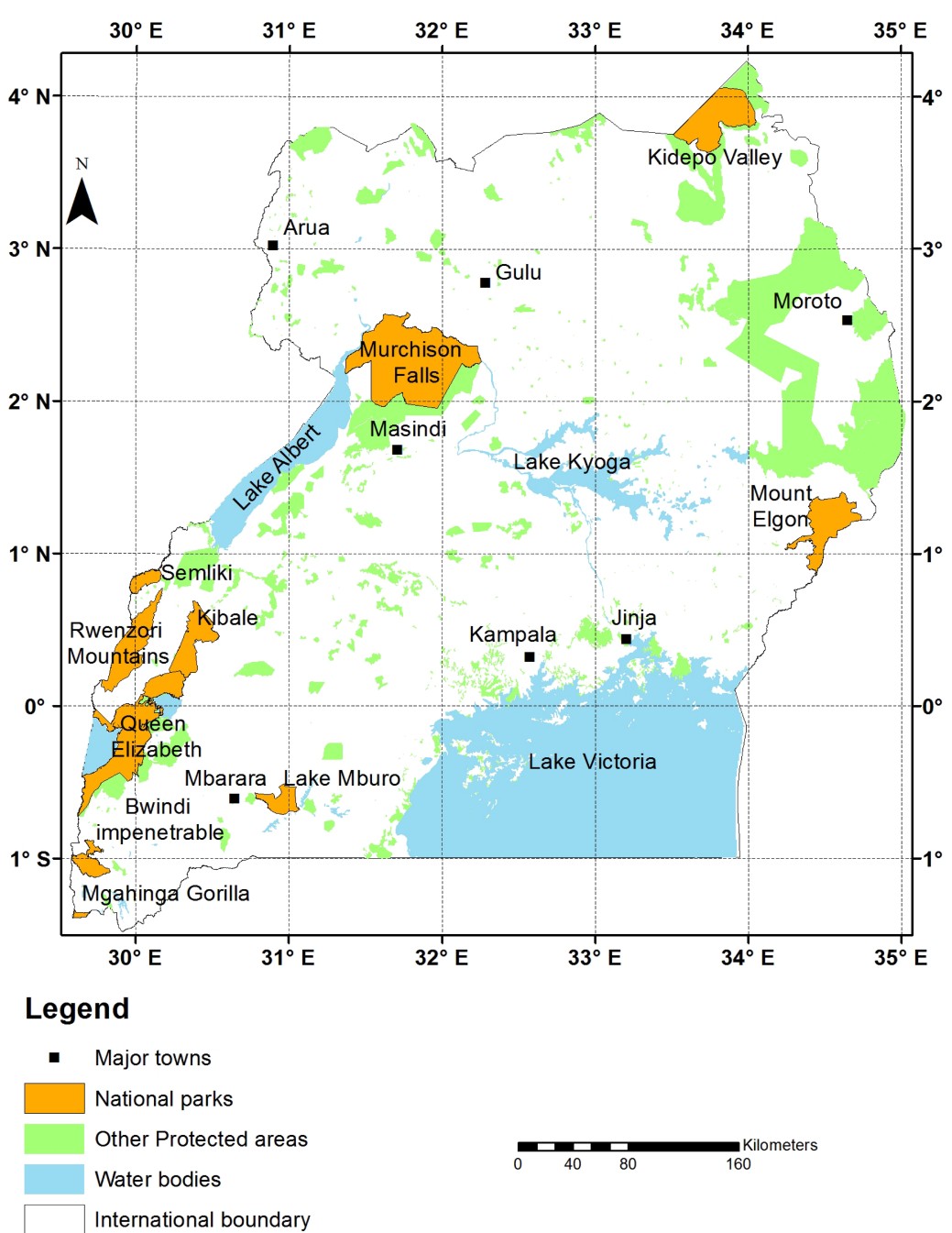

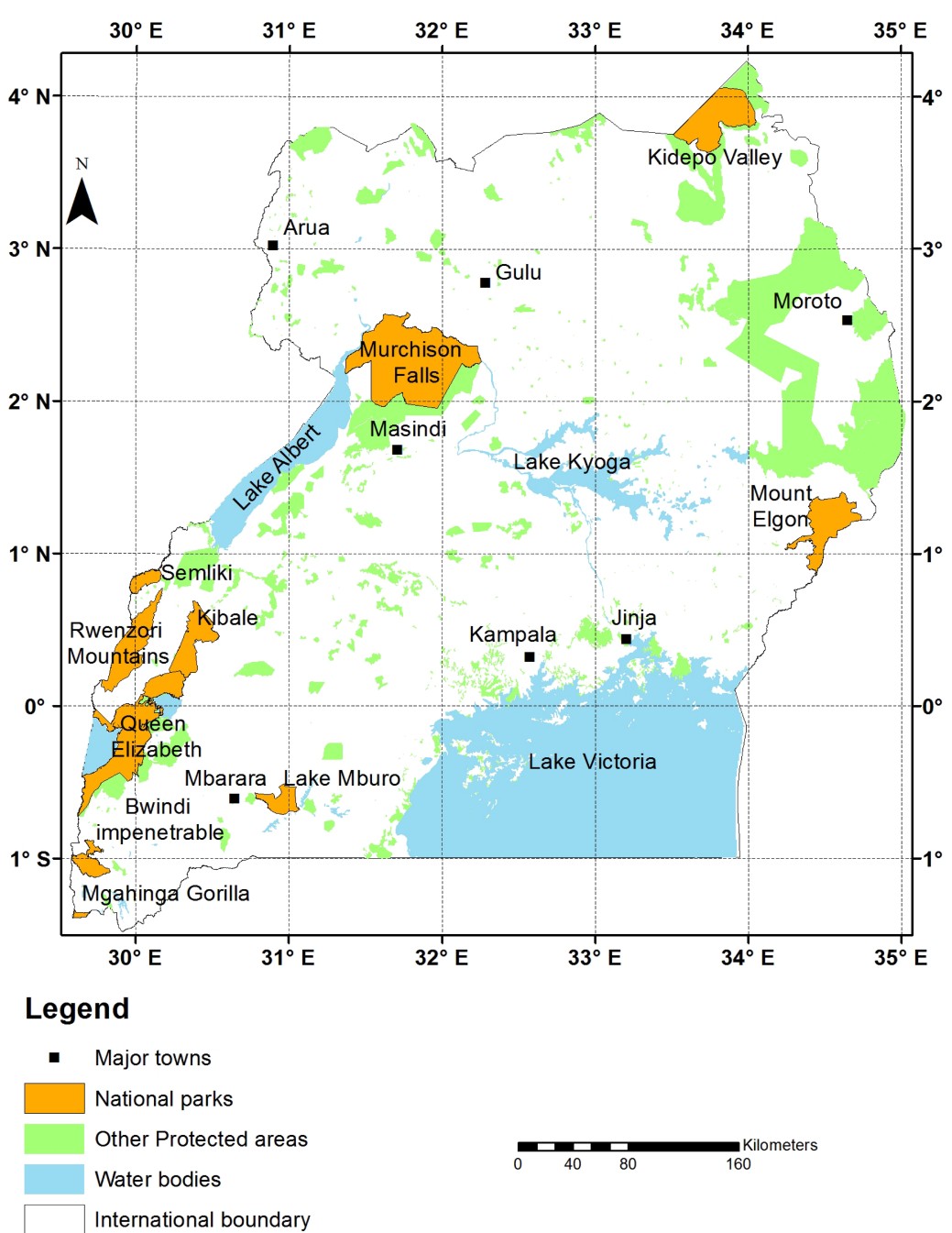

**Legend**

- ■ Major towns
- ▮ National parks
- ▮ Other Protected areas
- ▮ Water bodies
- ▯ International boundary

**Figure 1 Map of Uganda.** Map of Uganda, Africa, showing protected areas, major lakes, and other landmarks adapted from *Behangana & Hughes (2019)*. Map was prepared by Grace Nangendo and Sam Ayebare of the Wildlife Conservation Society.

such as reassigning species that were in listed in an incorrect genus (*e.g.*, *Chamaeleo carpenteri* is actually in the genus *Kinyongia*), removing duplicated species listed in different genera (*e.g.*, *Ameitophrynus kisoloensis* and *Sclerophrys kisoloensis*), and combining

species that were listed multiple times—often due to misspellings of the specific epithet. Three sources (VertNet, iDigBio, and GBIF) included subspecies in their official lists for Uganda. For species that also had subspecies within these three lists, we treated different trinomials as separate species to reflect the diversity as represented within each list (*e.g.*, *Hyperolius viridiflavus shubotzi* and *Hyperolius viridiflavus pachyderma*). We elected to analzye trinomials this way for three important reasons. First, the IUCN Red List of Threatened Species includes subspecies as entities in need of a level of protection distinct from the species as a whole, thus recognizing subspecies is aligned with the major goals of the IUCN. Second, our approach is consistent with the updated concept of subspecies and its use in herpetology proposed by *De Queiroz (2020)* where trinomials are a "representational device that can (but need not) be used to indicate the nesting of incompletely separated lineages within a more inclusive lineage". Third, the different sources consulted included subspecies as distinct entries, and as a result, practitioners interested in quantifying regional diversity would, in most cases, simply use the total diversity as recognized by the source chosen. Nevertheless, in cases when a species was listed with a nominate subspecies (*e.g.*, *Dendroaspis polylepis* and *Dendroaspis polylepis polylepis*), we combined them into a single species to minimize overinflating diversity. These procedures to correct within-list issues were done in an attempt to ensure greater accuracy of species diversity as recognized by the different sources consulted.

We made quantitative comparisons by assessing the species richness and composition among all the cleaned and compiled lists. We used the IUCN Red List of Threatened Species to compare the number of species in each threatened category among the lists. To examine species composition among lists, we compared sources expected to be the most comprehensive for the country, such as synoptic guides for each group written by taxonomic authorities (*e.g.*, *Channing & Rödel, 2019*; *Spawls et al., 2018*), continually updated academic websites managed by experts (*e.g.*, ASW; AmphibiaWeb; The Reptile Database; RepFocus), and the IUCN Red List of Threatened Species accounts of which are written by scientists. We used the R package *ggVennDiagram* (*Gao, Yu & Cai, 2021*) in Program R version 4.2.2 (*R Core Team, 2022*) with the R studio interface (*Posit Studio, 2022*) to compare species composition among the most comprehensive taxonomic sources for each group separately, thus, sources that relied on more opportunistic data were excluded from this particular analysis.

## RESULTS

Among all sources, amphibian species richness ranged 40 to 121 species with an average of 90 species (Table 1). Among the four most comprehensive sources, 48 species were shared across all of them and the number of species unique to each source ranged from three to 14 (Fig. 2). Of the species across the lists with any threatened status on the IUCN Red List, sources ranged from one to five species, while Data Deficient species ranged from one to six. Overall, the most differences across lists were species that were either not listed on the IUCN Red List (range = 1–22 species), or species listed as Least Concern (range = 39–101 species).

**Table 1** Number of amphibian species in Uganda according to several resources and the number of species that are listed in the conservation categories on the International Union for Conservation of Nature's Red List of Threatened Species.

| Source | Species | NL | LC | DD | NT | VU | EN | CR |
|---|---|---|---|---|---|---|---|---|
| *Channing & Rödel, 2019* | 88 | 2 | 77 | 4 | 1 | 2 | – | 2 |
| Amphibian Species of the World | 86 | 3 | 75 | 4 | 2 | 1 | – | 1 |
| AmphibiaWeb | 63 | 1 | 56 | 4 | – | 1 | – | 1 |
| iDigBio | 116 | 7 | 101 | 4 | – | 3 | – | 1 |
| IUCN | 84 | – | 78 | 3 | – | 2 | – | 1 |
| iNaturalist | 40 | – | 39 | 1 | – | – | – | – |
| GBIF | 119 | 14 | 97 | 6 | – | 1 | – | 1 |
| VertNet | 121 | 22 | 93 | 5 | – | 1 | – | – |

**Notes.**

NL, Not listed; LC, Least concern; DD, Data deficient; NT, Near threatened; VU, Vulnerable; EN, Endangered; CR, Critically endangered.

Among all sources, reptile species richness ranged 86 to 219 species with an average of 176 species (Table 2). Among the four most comprehensive sources, 128 species were shared across all of them, while the number of species unique to each source ranged from four to 13, and at least two sources had >10 species unique to each (Fig. 2). Of the species across the lists with any threatened status on the IUCN Red List, sources ranged from seven to 17 species, while Data Deficient species ranged from one to four. Overall, the most differences across lists were species that were either not listed on the IUCN Red List (range = 5–30 species), or species listed as Least Concern (range = 67–160 species).

## DISCUSSION

We assessed the herpetofauna of Uganda as portrayed on popular biodiversity repositories which revealed variation in species richness and composition among these ostensibly authoritative lists. Results indicated that these sources disagreed on the herpetofauna species that occur in this East African country. Essentially, a different conclusion would be reached on the number and composition of species that occur in Uganda depending upon which source was consulted, a finding that shows how the quality of biodiversity data can reflect legacies of social inequality that impact present-day funding and research infrastructure (*Chapman et al., 2024*). The persistence of outdated taxonomies, the use of old synonyms, and inclusion of long-abandoned subspecies in some repositories point to a broader challenge in maintaining accurate and updated taxonomic standardization in repositories of species information (*Smith et al., 2000*; *Hortal et al., 2015*). Our findings indicate a significant need for more surveys to carefully document species distributions and more taxonomic assessments to better resolve species boundaries in East Africa, results of which can be quickly adopted on widely available biodiversity platforms. Ultimately, we call for greater investments in capacity building for Uganda specifically and the Afrotropics generally so that such field surveys and genomic studies can be completed by the people that would benefit the most from improved knowledge of their biodiversity.

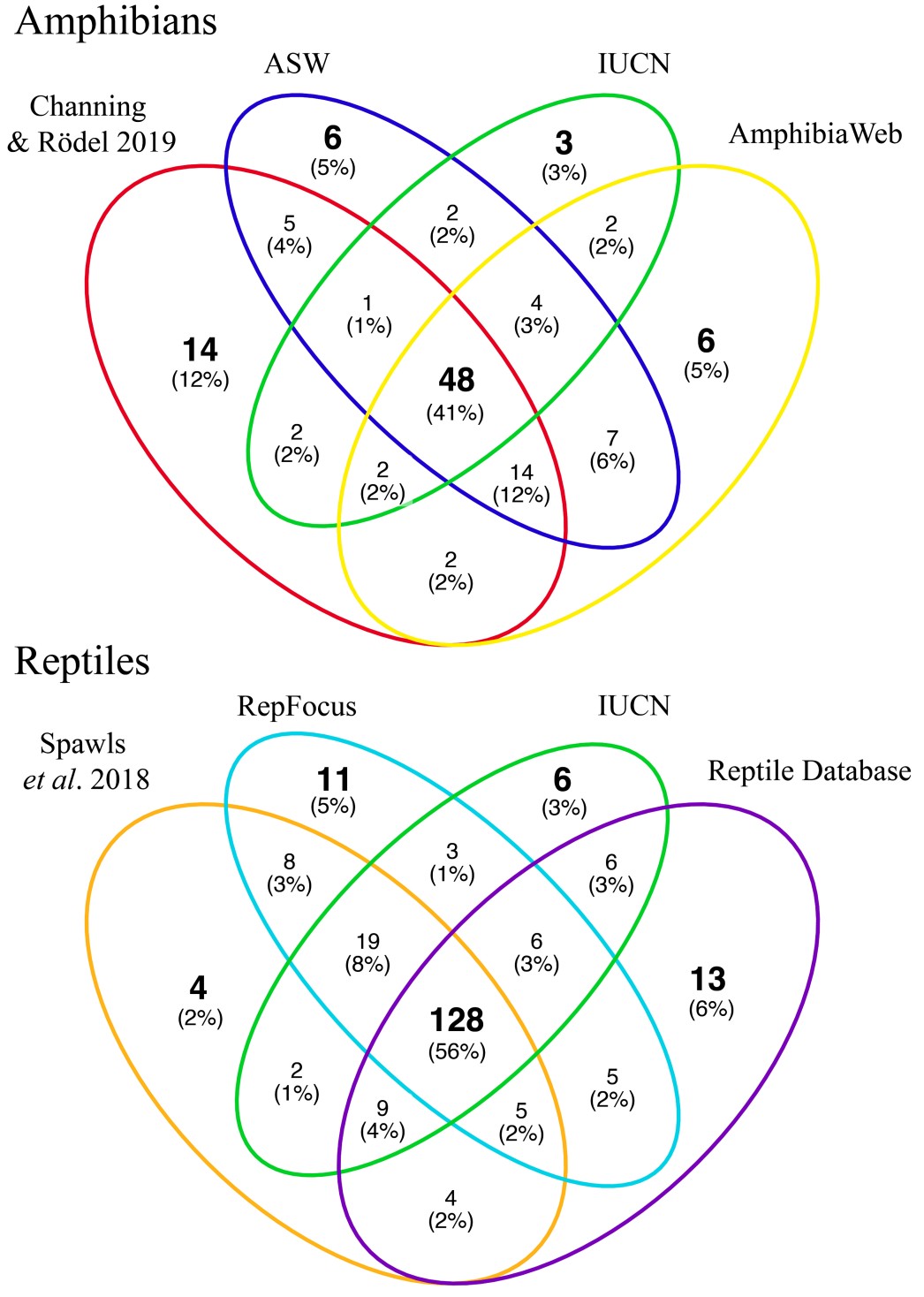

**Figure 2** **Species overlap as reported in major biodiversity sources for Uganda.** Venn diagrams showing the overlap in species composition among the four most comprehensive sources of biodiversity information for amphibians (top) and reptiles (bottom) in Uganda, Africa. IUCN, International Union for Conservation of Nature's Red List of Threatened Species; ASW, Amphibian Species of the World. See Methods for information on sources.

**Table 2  Number of reptile species in Uganda according to several resources and the number of species that are listed in the conservation categories on the International Union for Conservation of Nature's Red List of Threatened Species.**

| Source | Species | NL | LC | DD | NT | VU | EN | CR |
|---|---|---|---|---|---|---|---|---|
| *Spawls et al., 2018* | 178 | 17 | 142 | 2 | 7 | 8 | 2 | – |
| RepFocus | 185 | 16 | 150 | 3 | 7 | 7 | 2 | – |
| iDigBio | 190 | 16 | 155 | 4 | 6 | 7 | 2 | – |
| IUCN | 178 | – | 160 | 2 | 7 | 7 | 2 | – |
| Reptile Database | 175 | 5 | 151 | 3 | 7 | 7 | 2 | – |
| GBIF | 219 | 30 | 153 | 1 | 7 | 5 | 3 | – |
| iNaturalist | 86 | 11 | 67 | 1 | 3 | 4 | – | – |
| VertNet | 200 | 24 | 158 | 2 | 7 | 7 | 2 | – |

**Notes.**

NL, Not listed; LC, Least concern; DD, Data deficient; NT, Near threatened; VU, Vulnerable; EN, Endangered; CR, Critically endangered.

Our results highlighted a prevalent issue among these sources, where species richness was either overinflated or underestimated based on comparison to a mean value across sources and comparisons to the most authoritative sources. In particular, we found that the sources with the most inflated species diversity relative to the averages in each group were the three that included subspecies (GBIF, VertNet, and iDigBio). Conversely, sources with the lowest species diversity relative to the averages have inexplicable omissions but are at least more conservative estimates. Essentially, such sources which contain errors and unsupported inferences greatly overstate (or understate) our collective knowledge of the herpetofauna in Uganda, and thus risk leading researchers into adopting spurious accounts of species diversity for the region. Amphibians exhibited the most variation in species number and composition across different sources, which can be attributed to taxonomic instability of the group, synonym usage, and incomplete species inventories. These discrepancies underscore the ongoing challenges in African amphibian taxonomy (*Channing & Rödel, 2019*) and highlight the need for continued efforts to address such nomenclatural issues, especially those on widely available internet resources (*Rondinini et al., 2014*). Reptile species exhibited more consistency across sources compared to amphibians. The disparity in reptile species showcases that even in taxonomically stable groups it is challenging to compile comprehensive species lists (*Spawls et al., 2018*). Unexplained omissions among reptile lists may have resulted from gaps in research efforts or insufficient documentation of species occurrences on databases (*Tolley et al., 2016*; *Achieng et al., 2023*), emphasizing the importance of comprehensive field surveys and regular data validation of repositories.

Incomplete biodiversity knowledge can influence conservation priorities, affect resource allocation, and direct management strategies (*Stephenson & Stengel, 2020*). Differences among popular species lists have direct implications for conservation practitioners because distorted species inventories can lead to misallocations of funds and resources (*Leader-Williams & Albon, 1988*). By acknowledging and actively working to rectify these lists, we can enhance the reliability of biodiversity resources currently available, and, in turn, conservation organizations can make more informed decisions about resource allocations.

The high variability observed across lists indicates a need for increased investment in taxonomy, field surveys, and museum collections in Africa to enhance the accuracy of biodiversity repositories (*Siddig, 2019*). Addressing these challenges is critical for ensuring that conservation decisions are based on accurate assessments of species richness, ultimately contributing to more just and effective biodiversity conservation strategies in Africa and around the globe (*Chapman et al., 2024*).

## ACKNOWLEDGEMENTS

We thank the proprietors of the biodiversity platforms researched herein for their efforts to maintain such valuable resources for the conservation community, often with little or no support. We thank Lukwago Wilbur, Joseph Isingoma, Bob Katabazi, and Eli Greenbaum for their contributions to herpetological research in Uganda. We thank two anonymous reviewers for their comments that improved this manuscript.

### Funding

The authors received no funding for this work.

### Competing Interests

Daniel F. Hughes is an Academic Editor for PeerJ. Mathias Behangana is employed by NICE Planet Uganda Limited.

### Author Contributions

- Daniel F. Hughes conceived and designed the experiments, performed the experiments, analyzed the data, prepared figures and/or tables, authored or reviewed drafts of the article, and approved the final draft.
- Mathias Behangana conceived and designed the experiments, authored or reviewed drafts of the article, and approved the final draft.

### Data Availability

The cleaned species lists are available in the Supplemental Files.

### Supplemental Information

Supplemental information for this article can be found online at http://dx.doi.org/10.7717/peerj.18704#supplemental-information.

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
