# Peer review of "How many reptile and amphibian species are in Uganda, and why it matters for global biodiversity conservation"

_PeerJ, doi:10.7717/peerj.18704_

## Round 0.1 · original submission · Major Revisions

As you can see, reviewer 1 has raised some very serious concerns about your paper. Unless you can address the issues of how you retrieved the data and how taxonomic revisions may have altered the names of the species involved, I will not be able to accept this.

Reviewer 1 ·

Basic reporting

The authors conducted studies to create a comprehensive overview of herpetofauna in Uganda, highlighting their conservation statuses based on IUCN red list criteria. Although the article contributes knowledge on regional herpetofauna diversity, it is too descriptive in its current form. First, the authors mentioned that they consulted a range of sources with varying source materials to capture the most complete assessment of the herpetofauna of this country. Worrisomely, from the tables, I could only see one primary literature each for amphibians and reptiles, and many of the other inventories were based on global repositories, of which the taxonomic statuses of most of these species cannot be ascertained. What happened to other primary literature, etc? The use of GBIF, though it is essential, is something authors need to show how they were able to get clean records devoid of misidentification, out-of-the-exact species distribution ranges, etc. It is not gainsaying that most of these studies might have been based on morphological-based identification instead of a combined morphology and molecular approach. Thus, authors should be mindful and find ways to verify the exact taxonomic statuses of each of the identified species. One of the most plausible approaches will be to look at the geographic range of each species in the IUCN, AmphibiaWeb and Reptile database to confirm their geographic distributions. Again, how exhaustive is this literature search, and how did the authors ensure it was comprehensive? I would expect the authors to use different databases, e.g., PubMed, etc., and then the book search to assemble their pieces of literature. After that, they can use specific inclusion and exclusion criteria to decide which paper to include or exclude from this study. Again, authors can use software such as Vosviewer to analyze this literature, showing the trends of herpetological studies in Uganda and some other predicting variables (e.g., authors, methods used in analyses, etc.) in their paper. Other analyses could present an overview of the distribution, traits, and habitat association of these herpetofauna from Uganda to show the spatial variations in the distribution of these herpetofauna. Instead of presenting only the IUCN red list status for these herpetofauna, authors could look through the specific threats to these species in IUCN and then use a histogram or other analyses to present the threats to herpetofauna in Uganda. What about taxonomic inconsistencies and revisions? I expected authors to delve further into showing erroneously identified species and species needing detailed taxonomic clarifications. Again, there is a need to include a map of Uganda (a map of Africa as an insert) showing some representative herpetofauna, especially those endemic to Uganda.

Experimental design

Based on my suggestions above, the experimental design of this study needs to be reconstructed to ensure this study is of high technical standard.

Validity of the findings

The validity of the findings are not robust and statistically sound. It is difficult to assess if species included in this study were accurately identified. The authors failed to show how they were able to address numerous misidentification scenario that is common with African herpetofauna due to the cryptic species arising from phenotypic plasticity and the use of morphology alone in species identification

Additional comments

None

Reviewer 2 ·

Basic reporting

This article uses clear and professional English. The literature cited is sufficient and has all necessary components. I would like to also see a figure of a map of Uganda, along with perhaps protected areas or potential biodiversity hotspots. Just something to make this feel more central to Uganda. This is not a hypothesis driven study, but instead is observational of challenges faced by biodiversity and conservation practitioners.

Experimental design

This is almost a meta analysis of sources on Ugandan herpetofauna. The question is clear and well defined and the correct sources were surveyed. The methods were sufficient.

Validity of the findings

All underlying data have been provided (supplemental file). Analyses are simple comparisons of lists. A recommendation to see if subspecies status mattered is added in section 4 below. The results form the conservation status questions were somewhat confusing, and could be clarified.

Additional comments

Minor, in-line comments are below

abstract
14: …”which is often…” This addendum makes the whole sentence overly complex and confusing. Please break up this sentence for clarity.
16: change “on the number” to “of the number”
17: “reflect differences in the source referenced” -> needs to be clearer what “source referenced” means. And again, this is an overly complex run-on sentence.
25: There is a word missing after “especially”. Maybe “in”
26: “We discuss” -> another overly complex and confusing run-on sentence.

Intro
Line 64: delete redundant “red list”
End of intro: this could end more constructively, saying that your work highlights that certain groups are most at risk for being underrecognized (often little brown frogs… who knows how many species there are), and you can also highlight what kinds of work would help: genetic studies delimiting poorly resolved species, increased periodic species assessments on IUCN redlist, etc.

Methods
92: Can also point to table 1 as it lists these sources too.

Results
A specific discussion of how the subspecies play into the differences in species number is important in the results. Like “of potential lineages, subspecies make up ¼. Of the lineages that are overrepresented or exclusive to the GBIF or iDigBio datasets, these subspecies make up 80%” (something like that so that we can tell if it’s mainly a difference of whether they count subspecies or not)
148: I don’t know what it means when it says “, and the most differences were among species either not listed in the IUCN Red List (range = 1–22 species), or those listed as Least Concern (range = 39–101 species)”. Differences in what? In their occurrence in the country? The part about the threatened species is confusing.

Discussion
164: missing parenthesis.
Overall: we really need 2 things. We need more surveys, and we need careful taxonomic (and genomic) assessments of species clades to better resolve if there are multiple species or not. If these are done, then Amphibian Species of the World is fast to update subspecies or new species statuses, which then get adopted into other platforms. There isn’t a lot of grant funding available for these surveys and genomic studies, so your discussion should call for that investment as well. And should call for more capacity building in the Afrotropics so that surveys can be more effective (when local people can visit multiple times and get a good sense of if a species is ever found there or just easily missed. The way the discussion is phrased now, it mainly sounds like people just have to resolve the lists, which is impossible without better data.

Supplemental
Some of the genus names are not capitalized.

---

## Round 0.2 · accepted · Accept

Thank you so much for revision your manuscript. We are happy to accept it!

Reviewer 1 ·

Basic reporting

The manuscript is well-improved, and the lexical structure is clear.

Experimental design

The experimental design is sound and rigorous

Validity of the findings

The conclusion are well stated and their datasets were well discussed.

Additional comments

The manuscript is now well-improved and I commend the authors for addressing all the raised comments.